# Cytokine-Induced Guanylate Binding Protein 1 (GBP1) Release from Human Ovarian Cancer Cells

**DOI:** 10.3390/cancers12020488

**Published:** 2020-02-19

**Authors:** Grazia Carbotti, Andrea Petretto, Elisabeth Naschberger, Michael Stürzl, Stefania Martini, Maria Cristina Mingari, Gilberto Filaci, Silvano Ferrini, Marina Fabbi

**Affiliations:** 1IRCCS Ospedale Policlinico San Martino, Biotherapies Unit, Largo R. Benzi 10, 16132 Genoa, Italy; graziacarbotti@gmail.com (G.C.); gfilaci@unige.it (G.F.); 2Core Facilities—Clinical Proteomics and Metabolomics, IRCCS Istituto Giannina Gaslini, Via Gerolamo Gaslini 5, 16147 Genoa, Italy; andreapetretto@gaslini.org; 3Division of Molecular and Experimental Surgery, University Medical Center Erlangen, Friedrich-Alexander-University Erlangen-Nürnberg, Schwabachanlage 12, 91054 Erlangen, Germany; Elisabeth.Naschberger@uk-erlangen.de (E.N.); Michael.Stuerzl@uk-erlangen.de (M.S.); 4IRCCS Ospedale Policlinico San Martino, Immunology Unit, Largo R. Benzi 10, 16132 Genoa, Italy; stefania.martini@hsanmartino.it (S.M.); mariacristina.mingari@unige.it (M.C.M.); 5Department of Experimental Medicine and Centre of Excellence for Biomedical Research, University of Genoa, Via L.B. Alberti 2, 16132 Genoa, Italy; 6Centre of Excellence for Biomedical Research and Department of Internal Medicine, University of Genoa, Via De Toni 14, 16132 Genoa, Italy

**Keywords:** ovarian cancer, interleukin-27, interferon-γ, secretome, guanylate binding protein 1, microenvironment

## Abstract

We showed that IL-27 shares several effects with IFN-γ in human cancer cells. To identify novel extracellular mediators, potentially involved in epithelial ovarian cancer (EOC) biology, we analyzed the effect of IL-27 or IFN-γ on the secretome of cultured EOC cells by mass-spectrometry (nano-UHPLC-MS/MS). IL-27 and IFN-γ modulate the release of a limited fraction of proteins among those induced in the whole cell. We focused our attention on GBP1, a guanylate-binding protein and GTPase, which mediates several biological activities of IFNs. Cytokine treatment induced GBP1, 2, and 5 expressions in EOC cells, but only GBP1 was secreted. ELISA and immunoblotting showed that cytokine-stimulated EOC cells release full-length GBP1 in vitro, through non-classical pathways, not involving microvesicles. Importantly, full-length GBP1 accumulates in the ascites of most EOC patients and ex-vivo EOC cells show constitutive tyrosine-phosphorylated STAT1/3 proteins and GBP1 expression, supporting a role for Signal Transducer And Activator Of Transcription (STAT)-activating cytokines in vivo. High GBP1 gene expression correlates with better overall survival in the TCGA (The Cancer Genome Atlas) dataset of EOC. In addition, GBP1 transfection partially reduced EOC cell viability in an MTT assay. Our data show for the first time that cytokine-stimulated tumor cells release soluble GBP1 in vitro and in vivo and suggest that GBP1 may have anti-tumor effects in EOC.

## 1. Introduction

Cytokines released by natural or adaptive immunity cells may act on the different cell types of the tumor microenvironment and contribute to shaping it [1,2]. For example, IFN-γ released by effector T cells exerts anti-tumor and anti-angiogenic effects but also mediates “adaptive immune resistance” through the induction of immune-suppressive circuits [3,4]. In particular, IFN-γ mediates PD-L1 expression on the tumor cells, which become resistant to the activity of Cytotoxic T-lymphocytes (CTLs) [5]. IFN-γ is also a potent inducer of the enzyme IDO1, which may further suppress anti-tumor T cell functions [6]. We found that IL-27, a cytokine of the IL-12 family [7,8,9], produced by myelomonocytic cells, shares several effects with IFN-γ such as the induction of IDO1, PD-L1 [10], and IL-18BP [11], in human epithelial ovarian cancer (EOC) cells. Indeed, a proteomic analysis of IL-27-modulated cellular proteins showed a high concordance with IFN-γ-modulated proteins. For instance, IL-27 induced the expression of mediators of IFN signaling and anti-viral activities, such as IRF1 and 9, IFIT1, 2, and 3 and the GTPases GBP1, 2, 4, and 5, in human EOC SKOV3 cells [12]. In addition, IL-27 up-regulated the HLA-class I antigen-presentation machinery and proteasomal proteins and, accordingly, the expression of membrane HLA-class I/β2-microglobulin complexes in human EOC, non-small-cell lung cancer, neuroblastoma [12], and small-cell lung cancer cell lines [13]. Therefore, IL-27, similar to IFN-γ may facilitate CTL recognition by HLA-class I antigen presentation [12], and also exerts anti-proliferative effects by directly acting on human tumor cells, including, for example, EOC [12] and multiple myeloma cells [14]. In addition, IL-27 inhibits the epithelial to mesenchymal cell transition and the production of pro-angiogenic factors, through STAT1-dependent mechanisms, in human non-small lung cancer cells [15]. On the other hand, the induction of PD-L1 and IDO1 expression may dampen the anti-tumor CTL response, suggesting that IL-27 and IFN-γ may have dual effects in tumors [16,17].

The overlapping functional effects of IFN-γ and IL-27 are related to the common usage of the STAT1 signaling pathway by the receptor complexes, although IL-27 also mediated STAT3 tyrosine phosphorylation [18]. Interestingly, EOC cells enriched from ascites showed constitutive phosphorylation of STAT1 and STAT3 and expression of IL-18BP and IDO1, suggestive of an in vivo role of STAT-activating cytokines in the EOC tumor environment [10,11].

Here, we first analyzed the effect of IL-27 or IFN-γ on the secretome of the EOC cell line SKOV3 in culture, with the aim of identifying novel extracellular mediators potentially involved in EOC biology. Our present results show that IFN-γ and IL-27 modulate the secretion of a limited number of proteins among those modulated in the whole cells. We then focused our attention on GBP1, which is one of the most significantly IL-27- and IFN-γ-induced proteins, both at the cellular level and in the secretome. GBP1 is a Guanylate-Binding Protein, which has GTPase activity and has been involved in IFN-mediated anti-microbial [19,20], anti-angiogenic [21,22,23], and anti-tumor activities [24,25,26]. GBP1 expression has been associated with a prognostically positive Th1 immune response in colorectal cancer [24,27,28]. These data were confirmed by a comprehensive study of the Cancer Genome Atlas Consortium, showing that high GBP1 expression correlates with reduced colorectal carcinoma aggressiveness [29]. Accordingly, the reconstitution of GBP1 expression in the GBP1-negative CRC cell line DLD-1 inhibited cell proliferation, migration, and invasion [26]. Moreover, inducible expression of GBP1 in mouse mammary adenocarcinoma TS/A cells inhibited their in vivo growth in syngeneic mice, predominantly through inhibition of angiogenesis [26]. Nevertheless, GBP1 expression may correlate with aggressive behavior in other cancers, including esophageal carcinoma [30] or glioblastoma [31], or with radio- or chemo-resistant phenotypes in different tumor cell types, including ovarian and head and neck carcinoma [32,33,34]. Here, we show that IL-27 or IFN-γ stimulated the release of full-length GBP1 in vitro, through non-classical secretory pathways, in the human EOC cell lines SKOV3 and OVCAR5 and in primary culture A161. More importantly, soluble GBP1 accumulates at high levels in the ascites of EOC patients.

## 2. Results 

### 2.1. Proteomic Analysis of IL-27- or IFN-γ-Stimulated EOC Cells Shows an Overlapping Pattern of Secreted Proteins, among Which GBP1 is Highly Expressed 

To address the effects of IL-27 and IFN-γ on the secretome of EOC cells, we used high-resolution mass spectrometry to identify and quantify large-scale proteins. To this end, we analyzed the conditioned media of untreated or cytokine-treated SKOV3 cells. Although this cell line is classified as an atypical non-serous ovarian cancer [35], it has been widely used as an EOC model, and it responds to IL-27 and IFN-γ stimulation by regulating a common set of cellular proteins [12].

Data processing through the MaxQuant software identified a total of 1062 secreted proteins, of which 929 were quantified. Applying an Anova Test with an FDR <0.01 and a S0> 0.1, we selected 699 proteins significantly modulated in the different conditions. The results were plotted as a heatmap, constructed starting from the LFQ (Label Free Quantitation) protein intensity values, which were normalized with Z-Score (Figure 1). 

We observed a common pattern of modulated proteins between the secretome of IL-27 and IFN-γ treated cells, as evidenced by group 6, composed of 343 up-regulated proteins, and group 3, formed by 51 down-regulated proteins, corresponding to a 56% of agreement. The processes associated with these groups were mainly linked to activation of the immune system, secretion, and primary metabolism (Appendix A). The remaining 44% of proteins showed a discordant trend with 181 up-regulated proteins in IFN-γ (groups 1 and 5), and 124 proteins, up-regulated only by IL-27 treatment (groups 2 and 4). The mainly enriched processes of groups 2 and 4 concerned the reorganization of the extracellular matrix and exocytosis, whereas groups 1 and 5 were more related to protein synthesis and degradation (Appendix A).

In order to highlight the differences between IL-27 and IFN-γ compared to the unstimulated control, a t-test was performed on the 929 quantified proteins. The result was graphed using a volcano plot (Appendix A). Overall, IL-27 and IFN-γ modulated the expression of 766 proteins in the conditioned medium with an FDR of <0.05 and a S0 > 0.1. We then compared these results with an analogous experiment performed on cell lysates from IL-27- or IFN-γ-treated or untreated SKOV3 cells, which yielded 940 t-test significantly modulated cellular proteins in the same conditions (dataset PXD004419). The Venn diagram (Figure 2A) shows that the overlap between the secreted and cellular proteins was limited, with just 89 commonly modulated proteins. A fold-change heatmap of the 89 commonly modulated proteins is shown in Appendix A. These findings suggest that cytokine-induced changes in the extracellular proteins do not merely reflect the protein modulation in the cell lysate. This conclusion was even more evident if we compared all the statistically significant modulated proteins from the t-test cell lysate and the secretome experiment. The new selection was based on a combination of both fold change >4 and gene ontology, associated with the extracellular component only (Figure 2B). This analysis led to the global identification of 74 proteins for the cellular lysate, of which only 25 were shared with the secretome. This difference supports the idea that the secretome is not a simple reflection of cellular protein modulation but is the product of a specific protein selection mechanism.

Interestingly, the guanylate binding protein interferon-inducible (GBP) 1, 2, 4, and 5 were all significantly induced in the cell lysate (Table 1) but only GBP1 was up-regulated in the secretome of cytokine-stimulated EOC cells, where the other GBPs were not detected. Since GBP1 has been involved in tumor biology, here we focused on this protein. 

### 2.2. GBP1 is Released Predominantly As A Full-Length Molecule by Cytokine-Stimulated EOC Cells

We first confirmed that SKOV3 cells release soluble GBP1 upon stimulation with IL-27 or IFN-γ, by the use of a GBP1 ELISA assay, whereas constitutive GBP1 secretion was low (Figure 3A). Similarly, the OVCAR5 EOC cell line and a primary human EOC cell line, IST-A161 [12], showed increased secretion of GBP1 in response to IL-27 or IFN-γ stimulation (Figure 3A). 

Previous studies showed that GBP1 can be released both as a full-length 67 kDa molecule and as a 47 kDa fragment, generated by caspase-1 cleavage, in cell culture supernatants from IFN-γ-stimulated endothelial cells [36]. Therefore, we performed a Western blot analysis of soluble GBP1, immunoprecipitated from the supernatant of cytokine-stimulated SKOV3 cells using an anti-GBP1 rat mAb [37]. As shown in Figure 3B,C, we could detect only the full-length, 67 kDa form of GBP1 in cell supernatants and cell lysates from IL-27-stimulated EOC cells. Although the 47 kDa fragment may not be detectable due to sensitivity issues, our data indicated a predominant secretion of the full-length GBP1 form in the cytokine-stimulated EOC cells. Even if some variability in the levels of soluble GBP1 was evident, in experiments using SKOV3 cells and with different detection techniques, consistent GBP1 secretion was observed in response to IL-27. The higher variability in the IFN-γ response may reflect, at least in part, the different culture conditions used for the assays and cytokine stability. 

### 2.3. GBP1 is Expressed by EOC Tumors In Vivo and Accumulates in the Ascites

We then hypothesized that GBP1 may be expressed in human EOC cells in vivo. Preliminary, we verified GBP1 expression in the TCGA-OV RNA-seq dataset containing 373 human ovarian cancers [38]. These data indicated variable expression of the GBP1 gene in ovarian tumors and a correlation of high GBP1 gene expression levels with better 5-year survival, considering either best or median separation of the data (Figure 4A). Protein Atlas available online [39] immunohistochemistry analyses of GBP1 protein confirmed protein expression in EOC cells, in vivo (Appendix A). Since in vitro cultured EOC cell lines showed limited constitutive expression of GBP1 in the cytoplasm and as the secreted protein, it was then possible that STAT1-activating cytokines, such as IFN-γ or IL-27, may drive GBP1 expression in vivo. Indeed, tumor-cells enriched from the ascites of EOC patients displayed constitutively tyrosine-phosphorylated forms of STAT1 and STAT3 [10,11], and also constitutive GBP1 expression (Figure 4B and Appendix A). We then analyzed the concentrations of IFN-γ and IL-27 in the ascites samples included in this study (Appendix A) by a milliplex assay. This small cohort consisted predominantly of over-55 years old patients (70%) with stage III-IV (73%), grade 2–3 (66%), and serous histotype (57%). As shown in Figure 4C, the levels of IFN-γ ranged from 70 pg/mL to undetectable, while substantial levels of immune-reactive IL-27 were found in most samples (from 4014 to 105 pg/mL), suggesting a potential role of IL-27 in the EOC environment.

Collectively, these data prompted us to investigate whether GBP1 may accumulate in the peritoneal fluids of ovarian cancer and possibly in the sera. Indeed, GBP1 ELISA analysis showed that extracellular GBP1 accumulated in the ascites of most EOC patients (with a mean value of 807 ng/mL, range 0–2,920 ng/mL). GBP1 levels in ascites were significantly higher than those observed in sera from healthy donors and EOC patients (Figure 5A). Indeed, GBP1 levels were only occasionally high in the sera of EOC patients and overall GBP1 serum levels did not show significant differences with serum GBP1 from age and sex-matched healthy donors. A paired analysis performed on ascites and sera from the same patients showed that GBP1 preferentially accumulates at higher levels in the ascites (Figure 5B), further supporting local GBP1 release at the tumor site. A preliminary analysis of survival of patients, stratified on the basis of GBP1 median level in the ascites, suggested a better survival of patients with high GBP1, although statistical significance was not reached due to the limited number of cases with available follow-up (Appendix A).

### 2.4. The Full-Length, Soluble Molecular Form of GBP1 Accumulates in the Liquid Phase of EOC Ascites 

To gain information on the GBP1 molecular form present in the ascites, we performed immunoprecipitation experiments using an anti-GBP1 mAb on the ascites of EOC patients. Western blot analysis of immunoprecipitated molecules showed the 67 kDa full-length GBP1 molecule, which was particularly evident in the ascites with high GBP1, as assessed by ELISA (Figure 5C). The 47 kDa form of GBP1 produced by caspase-1 cleavage was undetectable by immunoprecipitation, also in ascites. 

GBP1 is a leaderless protein and should not be released through the classical protein secretion pathway but rather through unconventional pathways, among which microvesicles release [40]. In addition, a previous report indicated GBP1 as a component of mesothelioma exosomal signature [41]. Therefore, we tested whether the full-length GBP1 released from cytokine-stimulated EOC cells in vitro or present in the ascites could be associated with cell-derived microvesicles. To this end, we performed a fractionated centrifugation to obtain 3000, 10,000, and 100,000× *g* microvesicular fractions and vesicle-free supernatant from the ascites. As shown in the representative experiment of Figure 5D, the full-length GBP1 form could be detected in immunoprecipitated molecules from the liquid (supernatant, SN) fraction, but it was barely detectable in the exosome-enriched, 100,000×g microvesicular fraction. Similar results were obtained in two additional experiments performed on different ascites samples (Appendix A).Since GBP1 is a GTPase and binds the GTP substrate, we tested whether the extracellular GBP1 present in the ascites could retain this capability through the use of agarose-GTP beads for GBP1 capture. As shown in Figure 6A,B, the full-length GBP1 form was detected by Western blotting with mAb 1B1 among the ascites proteins eluted from the agarose-GTP beads. Different from GBP1, GBP2 and GBP5 were undetectable among the same agarose-GTP eluted ascites proteins (Figure 6B). Of note, all these GBPs were expressed upon IL-27 treatment in the cytoplasm of EOC cells (Figure 6C), but only GBP1 was released in the cell supernatant, in agreement with proteomics data (Table 1). A prominent 55-52 kDa band observed in Western blots from agarose-GTP pull-down experiments, was characterized as a human-IgG heavy chain contaminant (Figure 6A).

### 2.5. Effects of GBP1 Gene Transfer on EOC Cells In Vitro

IL-27 or IFN-γ induces GBP1 gene transcription [12], therefore, we asked whether induction of GBP1 gene expression by transfection of a GBP1 expression plasmid was sufficient to mediate GBP1 secretion. As shown in Figure 7, transient transfection with the GBP1-pcDNA3.1 expression plasmid resulted both in strong GBP1 intracellular protein expression (Figure 7A) and in high levels of soluble GBP1 released in the culture medium (Figure 7B). Since GBP1 expression mediates anti-tumor effects in different models [25,26,28], we asked whether GBP1 had inhibitory effects in EOC cell models in vitro. As shown in Figure 7C, GBP1 hyper-expression results in a low, albeit significant, inhibition on SKOV3 cell viability as detected by a MTT assay, with respect to mock transfection. A similar inhibitory effect of GBP1 gene transfection on cell viability was observed in A161 and IGROV1 cells (Figure 7C), but not in OVCAR5. As GBP1-transduced cells release soluble GBP1, we wondered whether soluble GBP1 could affect SKOV3 cell viability. However, transfer of GBP1-trasfected cells conditioned media to un-transfected cells had no effect, suggesting that the reduction of cell viability was related to GBP1 intracellular expression.

The densitometry readings or intensity ratio of the relevant bands for all Western blot figures, together with the whole blots for each figure panel, are shown in the Appendix A.

## 3. Discussion

Cytokines produced by cells of innate or adaptive immunity are important mediators of intercellular signals within the tumor microenvironment, where they regulate several processes including inflammation, tumor cell proliferation, and progression, and the anti-tumor immune response [1,2,3,4]. In addition, several soluble factors released by tumor cells play a relevant role in the regulation of these processes. Here, we studied how IFN-γ and IL-27, which modulate the expression of a broadly overlapping set of intracellular proteins in SKOV3 EOC cells [12], may modify the secretome of these cells. Our present data show that these cytokines induce the secretion of a partially overlapping set of proteins, among which GBP1, an important intracellular mediator of IFNs activities [20]. Of note, only a minor fraction of proteins that are modulated by IFN-γ or IL-27 at the intracellular level are concordantly modulated in the secretome, as in the case of GBP1. In particular, other GBP family members, whose expression levels are increased by cytokine stimulation in the whole cells, are not detectable in the secretome. In addition, only GBP1 but not GBP2 and GBP5 were detected in patients’ ascites, as suggested by agarose-GTP pull-down experiments. In addition, cytokine-stimulated endothelial cells release only GBP1 but not GBP2 and GBP5 (Naschberger E. and Stürzl M., personal communication). Intriguingly, GBP1 can heterodimerize with GBP2 and GBP5 intracellularly [42], thus opening the question of why GBP1 is selectively secreted in cytokine-stimulated EOC cells. These findings also indicate that GBP1 secretion does not simply reflect a general release of intracellular proteins related to cell damage or death. Indeed, although IFN-γ or IL-27 can induce EOC cell apoptosis, this occurs at later time points than those of supernatant harvesting [10]. The finding that GBP1 gene expression by transfection, in EOC cells, results in GBP1 secretion, further suggests that the mechanisms involved in GBP1 secretion are constitutively active in EOC cells. Importantly, we found very high GBP1 concentrations in the EOC ascites, relative to levels found in the sera from healthy controls or EOC patients. Therefore, high GBP1 concentrations in the ascites may reflect local production by EOC cells, as evidenced by GBP1 detection by immunohistochemistry analyses in EOC samples [39] and by Western blot analysis of tumor cell-enriched preparations from ascites.

Regarding the possible role of IL-27 and IFN-γ in the induction of GBP1 expression and release in EOC in vivo, it is of note that low levels of IFN-γ were present in the ascites, whereas, on average, about 100-fold higher concentrations of IL-27 were detected. While IFN-γ levels detected may be low due to its short half-life, the amount of IL-27 may be overestimated by the Luminex assay used. Although the capture and detection antibodies used were raised against the whole IL-27 heterodimer, a partial cross-reactivity with other cytokines sharing one of the two IL-27 chains (p28 and EBI3) [7,8,9], cannot be formally excluded. Whatever the role of IL-27, additional data indicate that STAT-activating cytokines drive GBP1 expression and secretion in the tumor environment. Indeed, while EOC cell lines such as SKOV3 and A2774 in vitro display tyrosine-phosphorylated STAT1 and STAT3 only upon cytokine stimulation [10], EOC cells enriched from ascites show constitutive STAT1 and STAT3 activation and GBP1 expression. Moreover, other STAT1-inducible proteins, such as IL-18BP [11] and IDO1 [10], were constitutively expressed by EOC cells enriched from ascites. Altogether, these observations suggest that IL-27 produced by myelomonocytic cells or IFN-γ released by T lymphocytes mediate STAT1 signaling and GBP1 expression and release by EOC cells in vivo. It must be considered that IL-27 is an inducer of Th1 responses [7] and IFN-γ is a Th1 cytokine also released by activated CTLs. Therefore, besides the possible anti-tumor effects of GBP1, the presence of soluble GBP1 could represent a potential marker of an underlying Th1 response in the tumor environment. It is tempting to speculate that the correlation of high GBP1 mRNA expression and better survival, could reflect an underlying anti-tumor T cell response. Indeed, a very recent study reported an association between somatic mutations of DNA damage repair genes and a specific immune signature, which included the GBP1 gene in EOC [43]. It is well established that defects in the DNA repair system are associated with increased mutational load and generation of tumor-associated neo-antigens, which enhance tumor immunogenicity and the potential responsiveness to immunotherapy [44]. 

A previous report showed that GBP1 could be secreted by endothelial cells but not by the tumor cell lines HeLa and HaCaT [45]. Here, we show for the first time that tumor cells, such as the EOC cell lines SKOV3 and OVCAR5 and the primary cells A161, can express and release GBP1, but only after IL-27 or IFN-γ stimulation. A subsequent study identified, together with full-length GBP1, a 47 kDa GBP1 form in the supernatants of IFN-γ-stimulated endothelial cells, and in CSF samples of patients with meningitis [36]. This 47 kDa GBP1 is a C-terminal fragment of the protein, generated by caspase-1 cleavage of the full-length GBP1, resulting in the removal of an N-terminal fragment. However, previous studies showed defective caspase-1 activity in a panel of EOC cell lines, as indicated by the release of a full-length, biologically inactive pro-IL-18 form, while the caspase-1-generated mature IL-18 was undetectable both in vitro [46] and in EOC ascites [47]. Indeed, another report showed that caspase-1 expression is down-regulated in EOC cells and its overexpression by gene transfer mediates apoptosis [48]. Accordingly, we detected only the full-length 67 kDa GBP1 as the predominant form in cytokine-stimulated SKOV3 EOC cells, their supernatants in vitro, and in EOC ascites. Nevertheless, the presence of the 47 kDa form, at levels below the sensitivity of the immunoprecipitation assay, cannot be formally excluded.

Since GBP1 is a leaderless protein, its extracellular release through the classical protein secretion pathway via the endoplasmic reticulum and Golgi may be ruled out. Several intracellular molecules are released by cancer cells as cargoes in different types of microvesicles [40], which include membrane blebs, exosomes, and larger vesicles. Therefore, we obtained different vesicular preparations and the corresponding liquid phases from EOC ascites by differential centrifugations and analyzed them for GBP1 by immunoprecipitation or ELISA. Our data indicate that the GBP1 67 kDa form is found predominantly in the liquid phase, supporting a non-classical mechanism of extracellular release but independent of microvesicles. In agreement with this observation, a previous report indicated that GBP1 release from endothelial cells can be inhibited by glyburide, further supporting a non-classical secretion pathway [45].

Regarding the possible functions of extracellular GBP1, we found that soluble GBP1 can be isolated by precipitation with GTP-agarose beads from ascites, indicating that it retains the ability to bind its substrate GTP. This finding suggests that GBP1 may participate in the regulation of the balance of extracellular guanosine nucleotides in the EOC extracellular environment, through its GTPase activity. Previous observations suggested that extracellular guanosine regulates extracellular adenosine levels [49] and may, therefore, modulate purinergic signaling pathways [50].

High GBP1 expression has been previously related to better [24] or worse prognosis [30] in different human tumors. In the transplantable breast cancer syngeneic model TS/A, GBP1 overexpression resulted in reduced angiogenesis and tumor growth [25]. Recent data indicated that the GBP1-mediated anti-proliferative effects on human colorectal cancer cells DLD1 are mediated by a specific amino acid sequence at the N-terminus of the GBP1-α9-helix, which interacts with the DNA-binding domain of the transcription factor TEA Domain protein (TEAD). In this way, GBP1 inhibits TEAD transcriptional activity and down-regulates TEAD-target genes [51]. The role of GBP1 in EOC is controversial. On the one hand, GBP1 over-expression has been associated with chemo-resistance [32,33], through a PIM1-mediated interaction [34]. In addition, Wadi et al., reported that higher GBP1 gene expression, detected by microarray technology, correlated with poorer progression-free survival, in a set of ovarian cancer patients with optimal tumor debulking and treated with platinum + taxol chemotherapy [52]. On the other hand, a more recent report included high GBP1 gene expression in a 7-gene functional interaction module that is associated with a better prognosis for EOC patients [53]. Similarly, we report that the level of GBP1 mRNA expression detected by RNA-seq correlates with better 5-year overall survival in the TCGA cohort consisting of 373 EOC samples, which include cystic, mucinous, and serous ovarian cancers [38]. These contradictory findings may relate to the different technical approaches used to evaluate mRNA expression levels (i.e., analysis based on a single affimetrix probe set versus RNA-seq) and/or use of different patient cohorts and subgroups, and endpoints (progression-free versus 5-year survival).

In the present study, we were unable to establish a correlation of soluble GBP1 protein levels and survival due to the limited consistency and follow-up of the cohort. Nonetheless, preliminary analysis suggests that further studies to evaluate the prognostic value of GBP1 levels in ascites are warranted. In addition, the potential role of soluble GBP1 as a biomarker of a Th1/CTL response in the EOC tumor environment should be considered.

## 4. Materials and Methods

### 4.1. Ethics Approval and Consent to Participate

This study was conducted in accordance with the ethical standards and according to the Declaration of Helsinki and National and International guidelines and has been approved by the authors’ Institutional Review Board (n.27/12). Blood samples and ascites were collected during surgical procedures from patients who gave written informed consent and used following the Institutional Review Board approval.

### 4.2. Cells and Cell Treatments

The human EOC cell lines SKOV3 (ATTC), OVCAR5, IGROV1, and IST-A161 [12] were grown in RPMI 1640, with L-glutamine, 10% FCS, and antibiotics (Lonza, Basel, Switzerland). A sample of each cell line was recently genotyped using the Cell IDTM System (G9500, Promega, Milan, Italy) and the GeneMapper^®^ software (ThermoFisher Scientific, Milan, Italy), version 4.0.

For secretome analysis, cells were seeded in 6-well plates at 1.5 × 10^5^ or 3 × 10^5^ cells/well in complete medium. The day after, the culture medium was replaced with medium containing 0.1% FCS with or without human recombinant: IFN-γ (1000 IU/mL, 300-02, PeproTech EC, London, UK) or IL-27 (100 ng/mL, 2526-IL-010, R&D Systems, Minneapolis, MN, USA). After 24 or 48 h, the supernatant was recovered, centrifuged at 2000× *g* to remove cell debris, and stored at −80°C until proteomic analysis.

### 4.3. Sample Preparation and Mass Spectrometer Setup

The cell supernatants were precipitated with acetone, and pellets were re-suspended in 25 µL of lysis buffer (6M GdmCl, 10 mM TCEP, 40 mM CAA, 100 mMTris pH8.5). The samples were reduced, alkylated, and lastly digested in a single step and then loaded into StageTip [54]. Peptides were analyzed by nano-UHPLC-MS/MS using an Ultimate 3000 RSLC with EASY spray column (75 μm × 500 mm, 2 μm particle size, Thermo Scientific, and with an 180-minute non-linear gradient of 5–45% solution B (80% CAN and 20% H_2_O, 5% DMSO, 0.1% FA) at a flow rate of 250 nL/min. Eluting peptides were analyzed using an Orbitrap Fusion Tribrid mass spectrometer (Thermo Scientific Instruments, Bremen, Germany). Orbitrap detection was used for both MS1 and MS2 measurements at resolving powers of 120 K and 30 K (at m/z 200), respectively. Data-dependent MS/MS analysis was performed in top speed mode with a 2 s cycle time, during which precursors detected within the range of m/z 375 to 1500 were selected for activation in order of abundance. Quadrupole isolation with a 1.4 m/z isolation window was used, and dynamic exclusion was enabled for 45 s. Automatic gain control targets were 2.5 × 10^5^ for MS1 and 5 × 10^4^ for MS2, with 50 and 60 ms maximum injection times, respectively. The signal intensity threshold for MS2 was 1 × 10^4^. HCD was performed using 30% normalized collision energy. One Microscan was used for both MS1 and MS2 events.

The mass spectra were subsequently processed using the MaxQuant software version 1.5.3.30 [55], and a human database downloaded from UniProt (release 2016_02). For all processing, the default settings have been maintained. Lastly, quantification was performed using the built-in label-free quantification algorithm [56], enabling the ‘Match between runs’ option (time window 1 min) in order to compensate for the effect of the missing values. Proteomic data have been deposited to the ProteomeXchange Consortium via the PRIDE partner repository [57] with the dataset identifier PXD016479 (submission details: Project Name: Cytokine-induced GBP1 release from human ovarian cancer cells; Project accession: PXD016479). All Statistics and bioinformatic analyses were performed with Perseus Software [58].

### 4.4. Western Blot

For the detection of intracellular GBP proteins, EOC cells were seeded in RPMI medium, the day after the medium was replaced and cytokines were added (IL-27 100 ng/mL and IFN-γ 1000 UI/mL). After 48 h of incubation, cells were detached by 2 mM EDTA solution in PBS, were lysed for 15 minutes in lysis buffer (20 mM Tris-HCl pH 7.4, 1 mM EDTA, 150 mM NaCl, 1% Brij97) containing 2 mM Na Orthovanadate and protease inhibitors (Complete Mini 04693124001, Roche Diagnostics, Mannheim, Germany). For the analysis of IL-27-induced tyrosine-phosphorylated STAT proteins, 2 × 10^5^ EOC cells were incubated for 20 min at 37 °C with or without 20 ng/mL of IL-27 in 0.1 mL of medium in a test tube. Cells were then rescued by centrifugation and immediately lysed.

Lysates were resolved under reducing conditions by 10% SDS-PAGE and analyzed by Western blotting using the following antibodies: Anti-GBP1 rat mAb 1B1 [37,59], rabbit anti-GBP2 (HPA042682), rabbit anti-GBP5 (HPA028656), rabbit anti-phospho-STAT1 (pY701), and anti-STAT1 anti-sera (9167 and 9172, respectively, Cell Signaling Technology, Leiden, The Netherlands), murine anti-phospho-STAT3 (pY705) and anti-STAT3 mAbs (612356 and 610190, respectively, BD Biosciences, San Jose, CA, USA), murine anti-Alix mAb (C-11 sc-271975, Santa Cruz Biotechnology, Heidelberg, Germany) and mouse α-tubulin mAb (T6074, Sigma-Aldrich/Merck, Milan, Italy). HRP-conjugated anti-mouse (P0447), anti-rabbit (P0448), and anti-rat (P0450) antisera were from Dako/Agilent Technologies (Milan, Italy); goat anti-human IgG (H10007) was from Caltag (ThermoFisher Scientific, Milan, Italy).

Proteins were detected by ECL Prime (RPN2232, GE Healthcare, Milan, Italy) and visualized by a chemiluminescence gel documentation and analysis system (MINI HD, UVITEC, Cambridge, UK).

Cells isolated from ascites were lysed in lysis buffer and analyzed by Western blotting, as above, in order to analyze in vivo the STAT3 or STAT1 phosphorylation status and GBP1 expression.

### 4.5. GBP1 Immunoprecipitation and GTP-Agarose Pull-Down

Anti-GBP1 1B1 rat monoclonal was coupled to Protein G-Sepharose beads (GE Healthcare, Milan, Italy) and cross-linked by bis(sulfosuccinimidyl)suberate BS3 (Pierce/ThermoFisher Scientific, Milan, Italy). Briefly, Protein G-Sepharose beads (400 μL) were washed with Hepes buffer (20 mM Hepes in 150 mM NaCl) and 1B1 antibody added for 1-hour incubation at room temperature, rotating at low speed. Unbound 1B1 was then washed out by centrifugation, volume adjusted to 1 mL with Hepes buffer and BS3 added to a final concentration of 2 mg/mL for an additional 1-hour incubation RT. The reaction was blocked by a 15 minutes incubation with Tris-HCl pH 7.5, 50 mM final concentration. Beads were then washed 5 times with PBS. Immunoprecipitation from ascites was carried out as follows: Ascites (1.5 mL/sample) were precleared from cell debris by centrifugation at 1000× *g*, diluted 1:1 with lysis buffer and supplemented with Protease Inhibitor Cocktail Set I (539131, Calbiochem/Merck, Milan, Italy,). Ascites were then precleared by 3 rounds of absorption on Protein G-Sepharose beads (1 hour RT), then challenged with anti-GBP1-Protein G-Sepharose beads (40 μL/sample, rotation at 4 °C overnight). Immunoprecipitates were then washed 5 times with lysis buffer and proteins eluted with Laemmli sample buffer, under non reducing conditions, boiled and loaded on a 10% acrylamide SDS-PAGE, followed by Western blot analysis.

For precipitation with GTP-Agarose (G9768, Sigma/Merck, Milan, Italy), ascites (1 mL/sample) were diluted with 3 mL of buffer containing 20 mM Tris-HCl pH 7.5, 150 mM NaCl, 10 mM MgCl_2_, 1 mM DTT, 1 mM PMSF, and 0.05% Triton-X-100. Precipitation with GTP-Agarose beads (100 μL/sample) was carried out overnight at 4 °C, rotating at low speed. Beads were then washed with the above buffer and proteins were eluted by boiling in reducing Laemmli sample buffer. SDS-PAGE and Western blot followed, as above.

### 4.6. Microvesicles Isolation from Ascites

To study the possible localization of GBP1 in the microvesicles and liquid phases of ascites, we fractionated them by centrifugation with a Beckmann Optima XPN-100 ultracentrifuge. Ascites were diluted 1:1 in PBS and centrifuged for 30 minutes at 3000× *g*. The supernatant was recovered and centrifuged for an additional 30 minutes at 10,000× *g*. The resulting liquid phase was centrifuged for 3 h at 100,000× *g* to obtain a microvesicle-containing pellet and a vesicle-free liquid phase. Pellets derived from each centrifugation were resuspended in PBS for analyses.

### 4.7. ELISA and Multiplex Immunoassay

To analyze soluble GBP1 release in cell culture medium, 3 × 10^5^ EOC cells were seeded in 6-well plate in complete RPMI medium; after 24 h, the medium was replaced with RPMI supplemented with 0.5% FCS and 100 ng/mL of IL-27 or 1000 UI/mL IFN-γ in duplicate wells. After an incubation of 48 h, 1.5 mL of supernatant from each well was centrifuged to eliminate cell debris and stored at −80 °C. Sera and ascites derived from EOC patients were assessed and compared with healthy donors’ sera. Microvesicles fractions obtained by centrifugation were disrupted by sonication prior to ELISA analysis. GBP1 ELISA was carried out as described [36,45].

IL-27 and IFN-γ levels in EOC ascites were simultaneously measured with a Bead-Based Multiplex Assays using the Luminex technology (MILLIPLEX MAP Human TH17 Magnetic Bead Panel, customized HTH17MAG-8K, Merck Millipore, Darmstadt, Germany), according to the manufacturer’s instructions.

### 4.8. Cell Transfection and MTT Assay

Empty vector pcDNA3.1 or GBP1-pcDNA3.1 vector [37] were transfected in SKOV3 cells using Lipofectamine 2000 Reagent (11668-019, Invitrogen/ThermoFisher Scientific, Milan, Italy) and serum-free Opti-MEM medium (51985-026, Gibco/ThermoFisher Scientific, Milan, Italy,), in 24-well plates. After overnight incubation, the medium was replaced with antibiotics-free, standard culture medium, and 48 h later, supernatants were collected for GBP1 ELISA assay.

To evaluate the effect of pcDNA3.1 or GBP1-pcDNA3.1 transfection on cell viability, we seeded 7 × 10^3^ SKOV3, A161 and IGROV1 cells/well in RPMI with 10% FCS in 96-well, flat-bottomed microtitre plates. After 24 h, cells were transfected as above. At 72 h from transfection, 20 µL of MTT (2 mg/mL in PBS) was added to each well for 4 h. Formazan salts were dissolved with 100 µL of DMSO/well and results were read on a spectrophotometer at 540 nm wavelength. Every condition was tested in 10 replicate wells. The assay was performed twice.

### 4.9. Statistical Analysis and Online Data Retrieval

The statistical analyses were performed using GraphPad PRISM version 5.03. The normal distribution of the data was verified before applying parametric tests. The one-way ANOVA and appropriate multiple comparison tests were used to compare expression levels between patients and control subjects. The paired Student t-test was used when appropriate. Non-parametric methods were used to examine the correlation between protein expression levels (Spearman’s correlation coefficient). *p* < 0.05 was considered significant for all statistical calculations. Values are given as mean ± SEM or mean ± SD.

The analysis of survival in relation to the GBP1 expression levels assessed in the TCGA-OV RNA-seq project was retrieved from https://www.proteinatlas.org/ENSG00000117228-GBP1/pathology/ovarian+cancer.

## 5. Conclusions

In this report, we show for the first time that the leaderless protein GBP1 is released by IL-27- or IFN-γ-stimulated ovarian cancer cells in vitro, whereas other members of the GBP family are induced intracellularly but not secreted. In addition, EOC cells express GBP1 protein and phosphorylated STAT1/3 proteins in vivo, suggesting a role for STAT-activating cytokines in GBP1 induction. Moreover, full-length GBP1 accumulates in patients’ ascites, suggestive of a potential role in the tumor microenvironment. High GBP1 gene expression correlates with better survival in the TCGA dataset of EOC patients, and transfection experiments suggest that GBP1 may affect EOC cell viability in vitro. Further studies to identify the mechanistic role of soluble GBP1 in the tumor environment of EOC and/or its role as a biomarker of an underlying immune response are warranted.

## Figures and Tables

**Figure 1 cancers-12-00488-f001:**
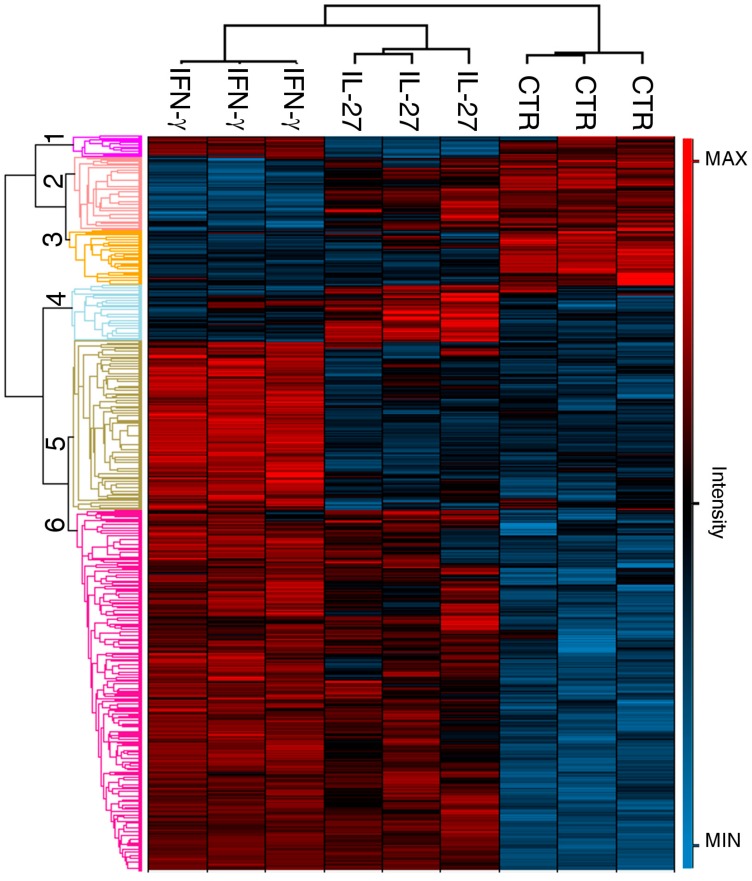
Heatmap of 699 cytokine-modulated proteins selected by multiple-samples test ANOVA, performed on the SKOV3 secretome. The graph (multiple-samples test ANOVA with FDR (False Discovery Rate) < 0.01 and S0>1, LFQ (Label Free Quantitation) intensity values normalized with Zscore) summarizes the effect on three independent biological replicates of conditioned media from cells treated with IL-27 or IFN-γ or left untreated (CTR). Proteins are clustered into six groups according to their expression value. Cluster #3 (51 down-regulated proteins) and cluster #6 (343 up-regulated) represent a concordant modulation by both cytokines. Clusters 1 and 5, 2 and 4 show a discordant trend with a total of 181 up-regulated proteins for IFN-γ and 124 proteins up-regulated in IL-27 treatment.

**Figure 2 cancers-12-00488-f002:**
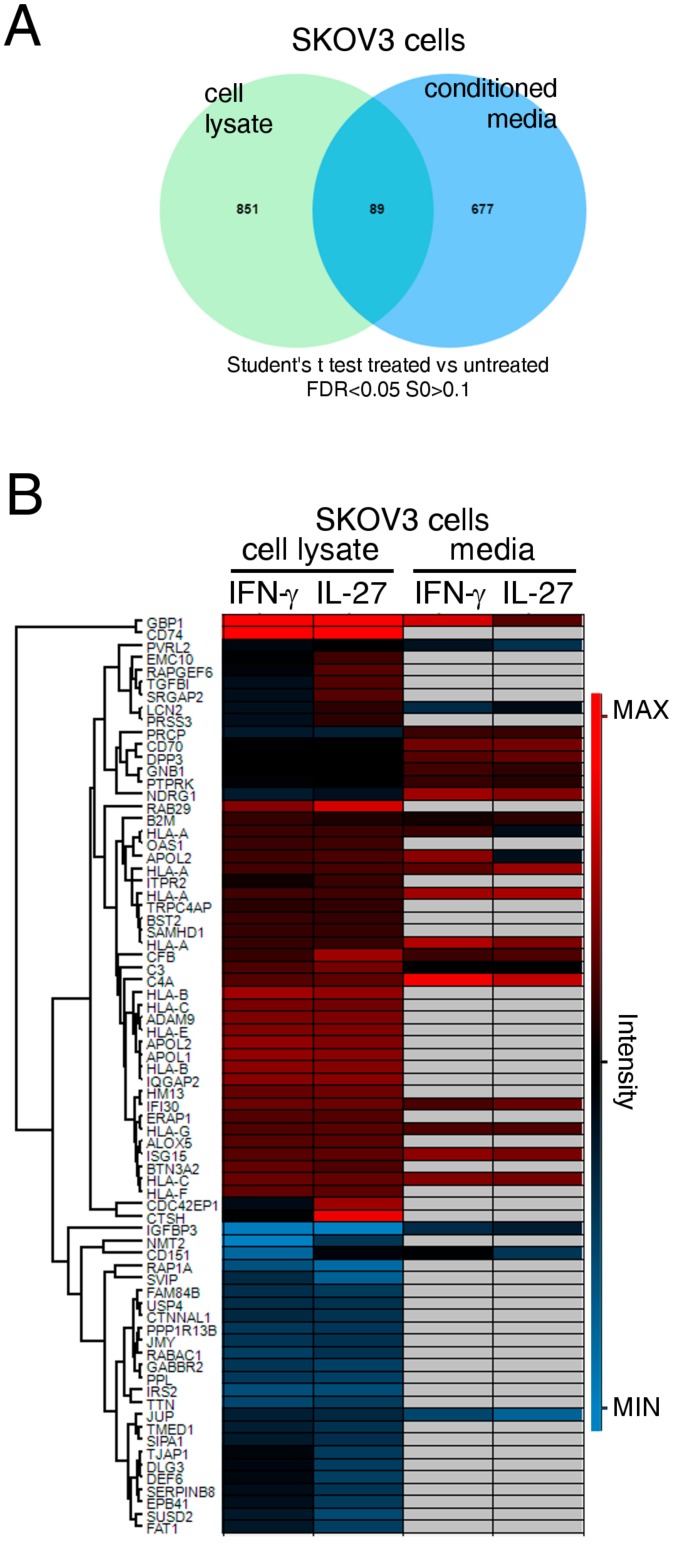
Comparison between the cellular proteome and the secretome of cytokine-stimulated SKOV3 cells. **A**) The Venn diagram (2**A**) shows the comparison between the t-test significantly modulated proteins (FDR < 0.05 S0 > 0.1 versus untreated) in cell lysates versus conditioned media of SKOV3 cells, under the same treatment conditions (IL-27 or IFN-γ). **B**) The fold-change heatmap shows the combination of further filtering t-test data applying a fold change >4, intersected with the gene ontology of the cellular component associated with the extracellular matrix. The new selection leads to the global identification of 74 cytokine-modulated proteins in the cell lysate, while only 25 are shared with the secretome.

**Figure 3 cancers-12-00488-f003:**
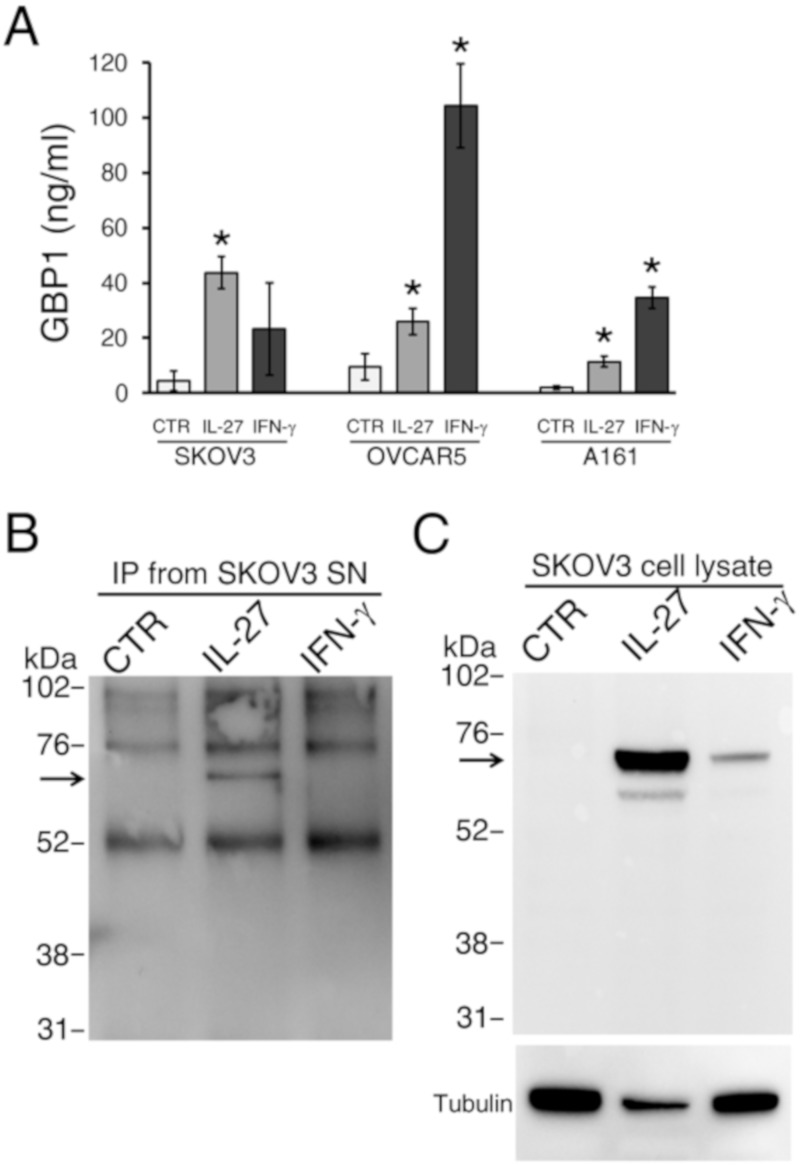
IL-27 or IFN-γ induces GBP1 secretion by EOC cell lines or primary culture, in vitro. **A**) A GBP1-specific ELISA assay shows increased levels of GBP1 in cell culture supernatants of IL-27 or IFN-γ-stimulated SKOV3 or OVCAR5 cell lines or A161 primary culture. Data for SKOV3 and OVCAR5 represent the mean ± SD of three different experiments (* *p* < 0.01). **B**) Western blot analysis of anti-GBP1-immunoprecipitated molecules from the supernatant (SN) of IL-27-stimulated SKOV3 cells shows an approximately 67 kDa band (arrow), which corresponds to the full-length GBP1 form. **C**) Western blot analysis of GBP1 in the corresponding cell lysates. Tubulin is shown as a loading control.

**Figure 4 cancers-12-00488-f004:**
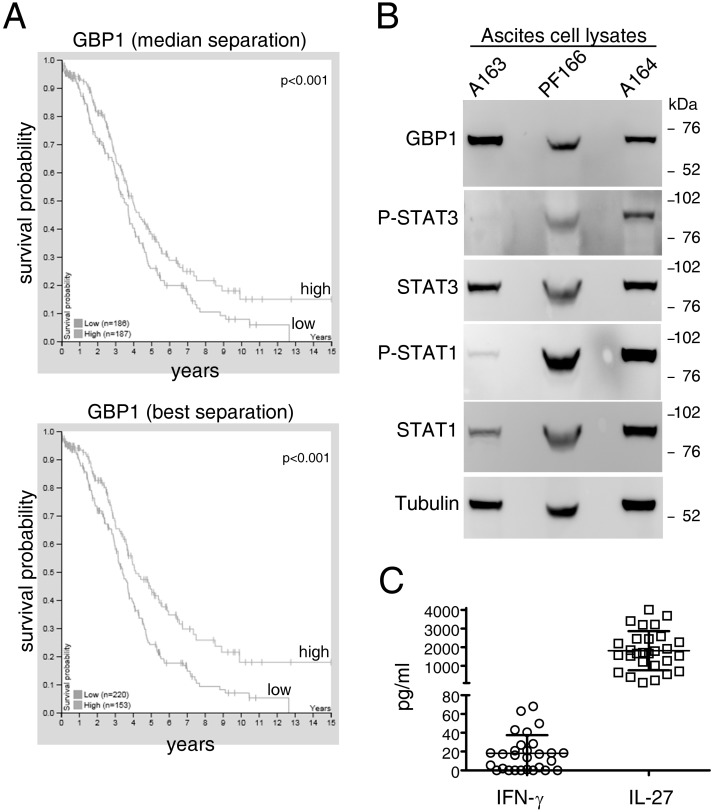
GBP1 and cytokine expression in epithelial ovarian cancer (EOC) tissues or ascites. **A**) Kaplan-Meier analysis of GBP1 expression in 375 epithelial ovarian cancers (EOCs) retrieved from the TCGA dataset (Human Protein Atlas): High GBP1 gene expression significantly correlates with better overall survival when using the median or the best value as the cut-off. **B**) Western blot analysis of tumor cell-enriched fractions from EOC ascites shows constitutive GBP1 expression and STAT1 and STAT3 tyrosine phosphorylation. Tubulin is shown as a loading control. **C**) IFN-γ and IL-27 levels in EOC ascites as detected by milliplex analysis.

**Figure 5 cancers-12-00488-f005:**
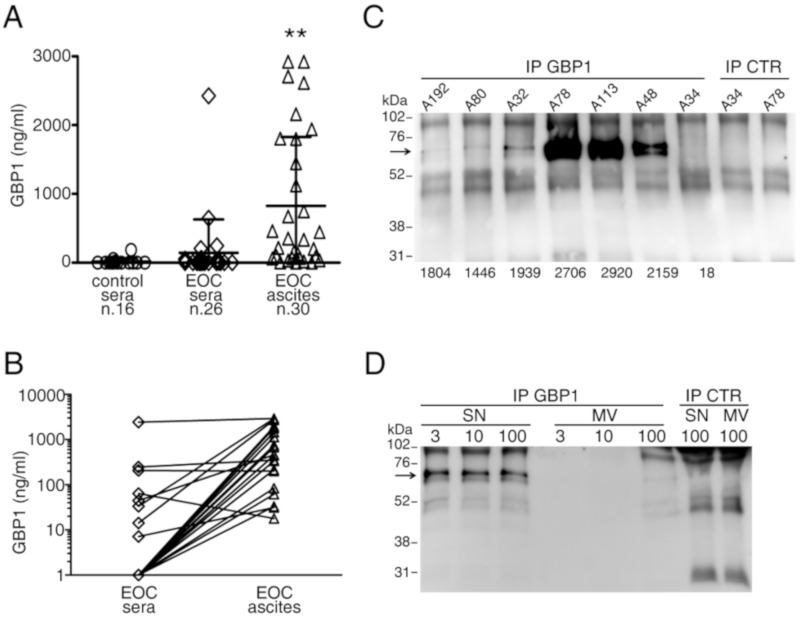
The full-length GBP1 protein form accumulates in the ascites of EOC patients. **A**) GBP1 levels, detected by ELISA, are significantly higher in the ascites of EOC than in the sera of EOC patients or healthy donors (** *p* < 0.002). **B**) A paired Student’s t-test analysis of sera and ascites from the same patients shows significantly higher GBP1 levels in the ascites (*p* < 0.0012). **C** )Western blot analysis of anti-GBP1-immunoprecipitated molecules shows a 67 kDa form in the ascites with high GBP1 but not in a low-GBP1 one. The numbers below each lane represent GBP1-ELISA values (ng/mL) of the samples. The last two lanes show a control IP using sepharose-protein G only. **D**) A high-GBP1 ELISA ascites (A101) was fractionated by centrifugation at 3000, 10,000 or 100,000× *g*. Anti-GBP1 immunoprecipitated molecules from the supernatant fractions or from the corresponding microvesicle fractions were analyzed by Western blot, which showed that the 67 kDa GBP1 is mostly in the supernatant fractions.

**Figure 6 cancers-12-00488-f006:**
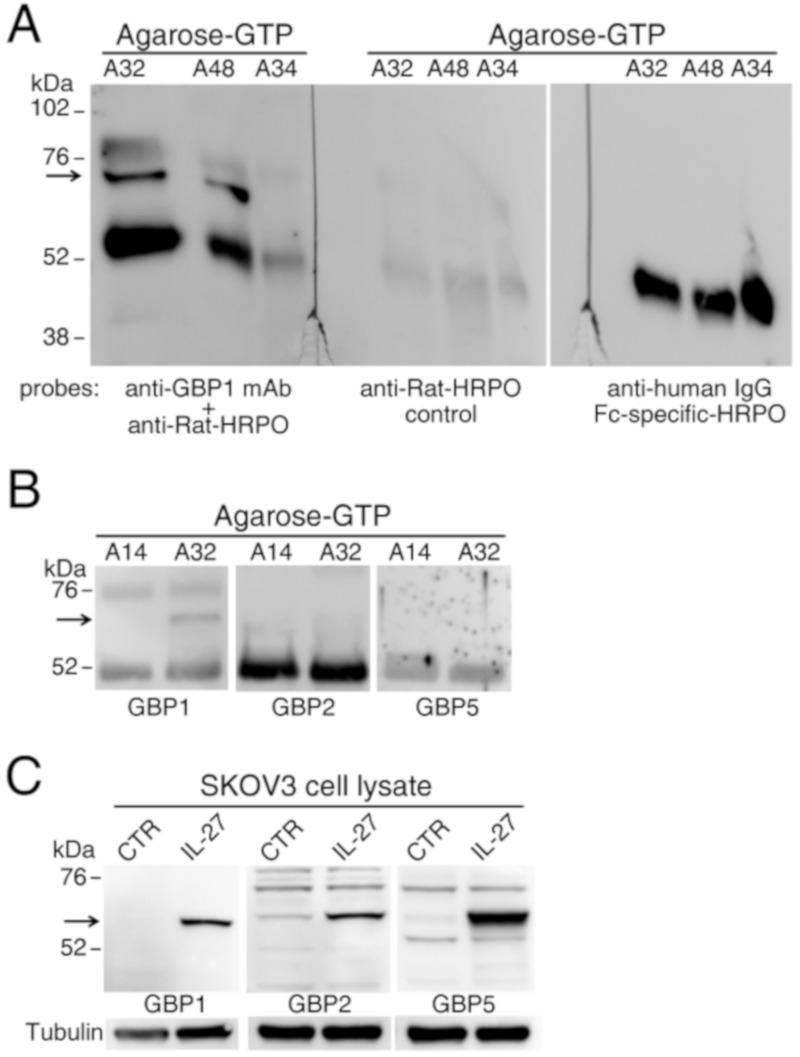
Enrichment of GBPs from EOC ascites by agarose-GTP beads shows the presence of soluble GBP1, whereas GBP2 and GBP5 were undetectable. **A**) The 67 kDa GBP1 molecule (arrow, left panel) is detected by Western blot in proteins enriched by agarose-GTP beads precipitation from the ascites with high GBP1 ELISA levels. Center panel shows a parallel blot probed with secondary antibody only. The same blot was re-probed with an anti-human IgG Fc-specific antibody (right panel). **B**) GBP2 and GBP5 were undetectable by Western blot of agarose-GTP-enriched ascites proteins, although (**C**) they were detected in cell lysates of IL-27-treated EOC cells, regarded as controls for antibody performance.

**Figure 7 cancers-12-00488-f007:**
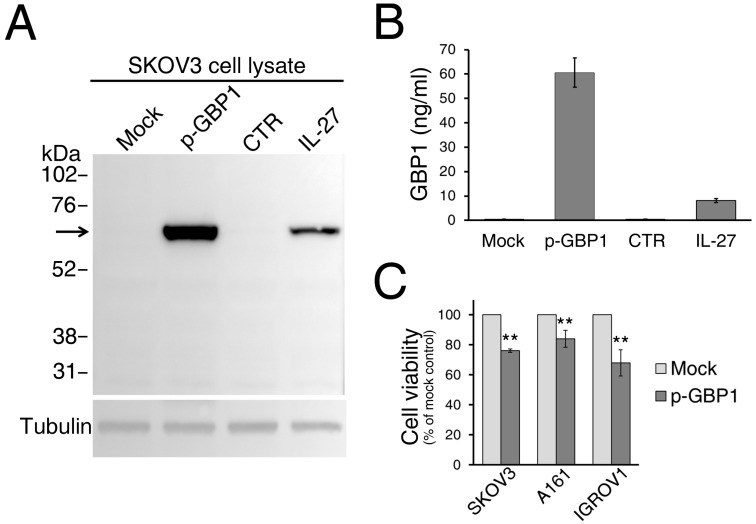
GBP1 gene transfer in EOC cells is sufficient to induce GBP1 secretion and results in decreased cell viability. **A**) Western blot analysis of GBP1 expression in cell lysates from SKOV3 cells transfected with vector control (mock) or GBP1-pcDNA3.1 plasmid (p-GBP1). Untreated (CTR) and IL-27-treated cells served as controls. **B**) GBP1 is detected in the conditioned medium of GBP1-pcDNA3.1-transfected cells and in IL-27-treated cells but not in controls. **C**) Transfection of GBP1-pcDNA3.1 reduces SKOV3, A161, and IGROV1 cell viability, assessed by MTT assay. Data are the mean of two independent experiments and are expressed as % of mock-transfected control cells. Error bars represent the minimum and maximum. ** *p* < 0.001.

**Table 1 cancers-12-00488-t001:** Comparison between the t-test significant fold increase in secretome and proteome of the cell lysate.

Protein	SKOV3 Cell Lysate		SKOV3 Conditioned Media	
	IFN-γ	IL-27	IFN-γ	IL-27
GBP1	8.50 ^a^	8.47	6.43	3.20
GBP2	5.95	6.18	NaN ^b^	NaN
GBP4	1.59	2.53	NaN	NaN
GBP5	5.73	6.78	NaN	NaN

a) Fold increase versus unstimulated control. b) Not detected.

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
