# Peer review of "Cytokine-Induced Guanylate Binding Protein 1 (GBP1) Release from Human Ovarian Cancer Cells"

_cancers, 2020, doi:10.3390/cancers12020488_

Round 1

Reviewer 1 Report

In this manuscript Carbotti et al. report original findings on the expression and role of the protein GBP1 secreted by ovarian cancer cells upon stimulation with IL-27 or IFNgamma. The study involves in vitro studies of proteomics analysis, and biological and biochemical experiments also performed with biological samples from patients (ovarian carcinoma and ascetic fluids). Overall the paper contains novel and interesting findings and the manuscript is very well written. I have no major criticisms, the experiments are well designed and well performed.

Just two notes:

In the Discussion, it would be of interest to the readers to expand a bit the possible relationship of GBP1 levels in the tumor/ascitic fluids and pattern of leukocyte infiltration/activation. As mentioned in the paper, GBP1 is a tumor-derived product resulting from the cross-talk with leukocytes producing IL-27 (or IFNgamma). Therefore some speculations can be advanced. The end sentence of the Discussion is unhappy: “..we were unable to establish a correlation….due to limited ..of the cohort study”. Besides the fact that they provide a TCGA analysis from a cohort with 373 patients - which is already a result- they can finish with a more optimistic sentence saying that they are currently pursuing studies to confirm the prognostic value of  GBP1 levels  in ovarian cancer patients.

Reviewer 2 Report

This study by G. Carbotti et al identifies a new role for IL-27 and IFN-g in the induction GBP1 from EOC patients. The paper is nicely written and the data are well presented.

The main issue I have is that it is not clear if the measurement of IL-27 in the multiplex assay detected the heterodimer of EBI3 and p28 or just one of the subunits. The statement in the discussion on lines 287-289 brings this into question. The authors should clarify how IL-27 was detected by the multiplex assay. If the dimeric form of the cytokine was not detected, then an IL-27 ELISA should be performed on the ascites.

Minor comments:

The sentence on line 64: “Our present results….cytokines in the whole cell” is hard to follow and would benefit from restructuring.

Line 99, 111, 130: please change INF to IFN

Reviewer 3 Report

The authors present evidence that GBP1 protein is secreted by ovarian cancer cells.  This observation followed work to analyze the changes in gene expression initiated by IL-27 treatment of epithelial ovarian cancer cells (EOCs).  This is an interesting study with both strengths and weaknesses.  Some of these will be listed below.

The introduction could be made more clear. Since GBP1 has been demonstrated to behave differently in different types of cancers (ex. Colon versus GBM), the cell lines used in the previous studies for which the results are described in the introduction should be specified. The authors frequently fall into just calling them cancer cells, which is too vague for this study.  The same is true for the discussion.

In table 1, the fold-increase in GBP1 in SKOV3 cell lysates is equal but there is twice as much GBP1 secreted by SKOV3 three cells when IFN-g treated versus IL-27 treatment. However this is not what is observed in Figure 3. In 3B, no GBP1 is IPed from supernatant of IFN-g treated cells and a much lower level is found in the cell lysate when compared to IL-27 treatment.  Either actin, GAPDH, or tubulin should be added to WB in 3C.  These results also conflict with the data from Fig. 2.  There the increase in GBP1 in media after IFN-g is much more than for IL-27.  These apparent discrepancies need to be addressed.

Would like to see more information for the materials and methods of the Kaplan Meier data. What year of data base with only 373 profiles? The data from which Affymetrix probe for GBP1 was used for the data analysis?  Were all ovarian cancers in that database screened or did that cover only a subset of ovarian cancer subtypes?  Wadi et al., showed that higher GBP1 expression in the 2016 TCGA database correlated with significantly poorer progression free survival.  These authors screened for all types of ovarian cancers in a database of 1648 ovarian cancers.  In addition, several papers have been published on GBP1 promoting Taxol resistance.  Since Taxol is part of the standard of care for ovarian cancers, GBP1 would be expected to decrease overall survival.  Again, please address these apparent contradictions a bit more.  The portion of the discussion related to these apparent discrepancies is really cursory. 

Figure 4B should have a loading control.

For Fig. 5, an exosome marker would be useful.

The agarose-GTP precipitations from ascites have a very prominent band at about 55 kD. Has this been identified?

For figure 7, cells were transfected with empty vector or GBP1 and shifted into antibiotic free media after 24 hours. After 48 hours, both the conditioned media and cells were collected. In part C, the authors make the case that GBP1 inhibits cell proliferation. In fact, their data does not directly address this.  There could be more cell death with transfection of the GBP1 containing plasmid than with the control vector.  Analysis of LDH release for cell lysis would be useful for most of these assays showing release of GBP1 into media, but particularly for this one.  A more direct measuring of proliferation would be necessary if the authors want to make the claim that GBP1 inhibits proliferation.

The fact that GBP1 had previously been shown to be induced in EOC by IL-27 and shown to be secreted by endothelial cells lessens the novelty/impact of this manuscript. It would be significantly improved by data showing that the CM from secreting cells could influence non-expressing EOCs.

Reviewer 4 Report

Authors provide a nice study on GBP1 and its role on cell proliferation.

Although its role in preclinical models is clearly described, I suggest to clarify the EOC patients in the Results section describing TableS1. 

Even if the number of patients analyzed is small (30 pts), I suggest to include in the result section just a sentence of an analyses of PFS  according to GBP1 expression reporting also median follow up (and a Kaplan-Meyer curve as supplementary figure).

Round 2

Reviewer 2 Report

I recommend acceptance of the manuscript.

Reviewer 3 Report

The authors have adequately addressed the reviewers comments.

Reviewer 4 Report

Authors clarified all the objections made.